# Longitudinal Changes in Body Composition of Long-Term Survivors of Pancreatic Head Cancer and Factors Affecting the Changes

**DOI:** 10.3390/jcm10153436

**Published:** 2021-08-02

**Authors:** Hyun-Ho Kong, Kyung-Won Kim, You-Sun Ko, Song-Cheol Kim, Jae-Hoon Lee, Ki-Byung Song, Dae-Wook Hwang, Won Kim

**Affiliations:** 1Department of Rehabilitation Medicine, Chungbuk National University Hospital, Cheongju 28644, Korea; jimlight@hanmail.net; 2Department of Radiology, Asan Image Metrics, Asan Medical Center, University of Ulsan College of Medicine, Seoul 05505, Korea; medimash@gmail.com (K.-W.K.); ko.yousun82@gmail.com (Y.-S.K.); 3Division of Hepatobiliary and Pancreatic Surgery, Department of Surgery, Asan Medical Center, University of Ulsan College of Medicine, Seoul 05505, Korea; drksc@amc.seoul.kr (S.-C.K.); hbpsurgeon@gmail.com (J.-H.L.); mtsong21c@naver.com (K.-B.S.); 4Department of Rehabilitation Medicine, Asan Medical Center, University of Ulsan College of Medicine, Seoul 05505, Korea

**Keywords:** pancreatic neoplasms, surgery, body composition, sarcopenia, recurrence

## Abstract

Previous studies on changes in body composition of pancreatic cancer patients have only focused on short-term survivors. We studied longitudinal body composition changes and factors affecting them in long-term survivors by analyzing many abdominal computed tomography images using artificial intelligence technology. Of 302 patients who survived for >36 months after surgery were analyzed. Multivariate logistic regression analysis for factors affecting body composition changes and repeated-measures analysis of variance to observe differences in the course of change according to each factor were performed. In logistic analysis, preoperative sarcopenia and recurrence were the main factors influencing body composition changes at 1 and 3 years after surgery, respectively. In changes of longitudinal body composition, the decrease in body composition was the greatest at 3–6 months postoperatively, and the preoperative status did not recover even 3 years after surgery. Especially, males showed a greater reduction in skeletal muscle (SKM) after surgery than females (*p* < 0.01). In addition, SKM (*p* < 0.001) and subcutaneous adipose tissue (*p* < 0.05) mass rapidly decreased in case of recurrence. In conclusion, long-term survivors of pancreatic cancer did not recover their preoperative body composition status, and preoperative sarcopenia and recurrence influenced body composition changes.

## 1. Introduction

The overall and recurrence-free survival of pancreatic cancer is increasing due to advances in diagnosis and treatment technology, but its prognosis is poor [1]. In addition to survival issues, approximately 70–80% of patients with pancreatic cancer develop cachexia, a significant loss of skeletal muscle (SKM) and adipose tissue (AT) mass, owing to increased systemic inflammation caused by factors such as activation of the catabolic pathway in the tumor itself [2]; problems after surgery such as exocrine pancreatic insufficiency, abdominal pain, and motility disturbance of the upper gastrointestinal tract [3]; and adjuvant treatments such as chemotherapy [4].

Patients with pancreatic cancer undergo body composition changes during their survival period. Sebastiano et al. reported that SKM and visceral AT (VAT) mass decreased significantly in patients during the follow-up period. The change was greater in cases with diabetes and anemia at the time of diagnosis of pancreatic cancer [5]. In another study, Dalal et al. indicated that SKM mass decreased by 4% and VAT and subcutaneous AT (SAT) mass decreased by 15% and 11%, respectively, during adjuvant treatment of locally advanced pancreatic cancer. Further, the study reported that the higher the patient’s body mass index (BMI) and VAT index, the greater the loss of SKM mass [6].

However, previous studies on longitudinal changes in body composition of patients with pancreatic cancer have focused on short-term survivors who eventually died. No studies have been performed on long-term survivors. In addition, the longitudinal observation period in previous studies was too short to conduct analyses of each factor that influences changes in body composition, and the number of subjects was very small to conduct subgroup analysis [5,6]. Knowledge of changes in the body composition that occur in pancreatic cancer survivors and the factors that influence them will help us improve the prognosis of patients by providing interventions, such as nutritional support [7] and exercise [8], through the early detection of patients who are likely to develop cachexia.

In this study, we analyzed the changes in body composition during the patient’s survival period and factors affecting the changes by analyzing a large number of abdominal computed tomography (CT) images of patients who survived for >36 months after surgery for pancreatic head cancer using artificial intelligence (AI) technology.

## 2. Materials and Methods

### 2.1. Patients Selection and Study Design

This retrospective observational study included 379 patients (225 males and 154 females) who underwent curative surgery for pancreatic ductal adenocarcinoma at Asan Medical Center (Seoul, Korea) from February 2000 to May 2014 and survived for >36 months. Since exocrine pancreatic insufficiency [9,10], malabsorption of fat-soluble vitamins and micronutrients such as zinc and iron [11,12] that can influence the nutritional status of long-term survivors is different depending on the location of pancreatic cancer and its surgical method, only pancreatic head cancer patients were included for homogeneity of the study population. Among them, subjects who met the following criteria were excluded from the analysis: (i) no abdominal CT performed within 2 months before surgery, (ii) no CT image at 12 or 36 months after surgery, (iii) CT image analysis using AI technology was not possible. Therefore, 302 patients (181 males, 121 females; a total of 906 abdominal CT images) with abdominal CT images taken within 2 months before surgery and 12 and 36 months after surgery were analyzed for factors influencing changes in body composition during survival. In addition, 196 patients (118 males, 78 females; a total of 1764 abdominal CT images) were analyzed for changes in longitudinal body composition during survival (Figure 1). This study was approved by the institutional review board (IRB) for our institution (IRB number: 2018-1273).

### 2.2. Surgical Procedure and Adjuvant Treatments

Surgery was performed by six experienced surgeons from the Division of Hepatobiliary and Pancreatic Surgery using conventional pylorus-preserving or pylorus-resecting pancreaticoduodenectomy according to the subject’s condition. After the surgery, all patients were treated according to a standardized postoperative clinical pathway. Adjuvant chemotherapy was administered for patients with stage ≥ II considering the patient’s condition, and radiation therapy was added for R1 resection. In borderline resectable cancer cases, neoadjuvant chemotherapy was added.

### 2.3. Anthropometrics and Laboratory Data

The height and weight of the patients were obtained from the institutional electronic medical records. BMI was defined as weight divided by the square of the height. Preoperative (preOP) blood tests were performed to measure the serum albumin level, neutrophil-lymphocyte ratio (NLR), C-reactive protein (CRP) level, carbohydrate antigen 19-9 (CA 19-9) level, and carcinoembryonic antigen (CEA) level.

### 2.4. Analysis of CT Images and Body Composition

SKM, SAT, and VAT mass was measured from CT images using the AI software (AID-U^™^, iAID Inc., Seoul, Korea) developed using a fully convolutional network-based segmentation method. An experienced radiologist selected the inferior endplate of the L3 vertebra without knowing the patient’s clinical information. The selected CT images were automatically demarcated according to each body composition type and the total abdominal muscle area (TAMA, cm^2^), subcutaneous fat area (SFA, cm^2^), and visceral fat area (VFA, cm^2^) were measured accordingly (Figure 2). A single experienced radiologist checked the segmentation results on all images. The TAMA included the psoas, quadratus lumborum, transverse abdominis, rectus abdominis, internal and external obliques, and paraspinal muscles. The TAMA was bounded by predetermined thresholds (−29 to +190 Hounsfield units (HS)), while SFA and VFA were demarcated using fat tissue thresholds (−190 to −30 HS) [13].

Skeletal muscle index (SMI, cm^2^/m^2^) was defined as the TAMA divided by the square of the height, and the subcutaneous fat index (SFI, cm^2^/m^2^) and visceral fat index (VFI, cm^2^/m^2^) were calculated by dividing the SFA and VFA by the square of the height, respectively. Patients were considered to have sarcopenia if their SMI was <25% of the study population. Obesity was defined as a VFA of ≥103.8 cm^2^, the standard abdominal obesity of the Korean population [14]. To analyze the factors affecting the change in body composition for 1 year and 3 years postoperatively, the analysis was performed by dividing participants into two groups according to the upper 25% with a large percentage decrease in SKM, SAT, and VAT mass.
$$\%\;\mathrm{change}\;\mathrm{of}\;\mathrm{SKM}\;\left(\%\right)=\frac{\mathrm{postOP}\;\mathrm{TAMA}\;\left({\mathrm{cm}}^{2}\right)\;–\;\mathrm{preOP}\;\mathrm{TAMA}\;\left({\mathrm{cm}}^{2}\right)}{\mathrm{preOP}\;\mathrm{TAMA}\;\left({\mathrm{cm}}^{2}\right)}\times 100\%$$

### 2.5. Statistical Analysis

All data are expressed as mean ± standard deviation (SD) or number (proportion). The student’s *t*-test was used to compare the mean values of continuous variables. The Pearson chi-square test or Fischer’s exact test was used in comparing differences between categorical variables. A paired *t*-test was used to determine whether the difference in body composition before surgery and 36 months after surgery was statistically significant. In addition, univariate and multivariate logistic regression analyses were performed to analyze the factors affecting the change in body composition at 12 and 36 months after surgery. The independent variables with a *p*-value < 0.10 in the univariate analysis were entered into the multivariate logistic regression model. In addition, a one-way repeated-measures analysis of variance was performed to analyze whether changes in longitudinal body composition from before surgery to 36 months after surgery showed a different course according to each factor. All statistical analysis was performed using SPSS version 25.0 (IBM, Armonk, NY, USA). A *p*-value < 0.05 was considered statistically significant.

## 3. Results

### 3.1. Patient Characteristics

Data from 302 patients (181 males and 121 females) were used for the analysis. There were no statistically significant differences between sex in age, BMI, cancer stage, recurrence rate, mean recurrence date after surgery, adjuvant treatment. Furthermore, there were no significant differences in laboratory findings between males and females, except the serum albumin level (*p* < 0.05). However, in the body composition analysis, SMI (*p* < 0.001) and VFI (*p* < 0.05) were significantly higher in males than in females, and SFI (*p* < 0.001) was significantly higher in females than in males. In addition, obesity was significantly higher in males than in females (50.3% vs. 32.2%, *p* < 0.01, Table 1).

### 3.2. Changes of Body Composition during the Survival Period

The percentage change in body composition at 36 months after surgery was as follows: SMI (male vs. female; −7.0 ± 10.2% vs. −3.6 ± 9.2%), SFI (−16.6 ± 38.1% vs. −22.7 ± 32.5%), and VFI (−27.5 ± 39.1% vs. −19.9 ± 55.3%). All the body composition types demonstrated statistically significant decreases (*p* < 0.001). In particular, SMI decreased significantly in males than females (*p* < 0.01, Table 2).

### 3.3. Factors Affecting Changes in Body Composition after Surgery

In the multivariate logistic analysis of the changes in body composition for 1 year after surgery, the risk of a decrease in SAT mass (odds ratio (OR) 2.33, 95% confidence interval (CI) 1.30–4.17, *p* < 0.01) and VAT mass (OR 2.05, 95% CI 1.15–3.65, *p* < 0.05) was higher, and the risk of a decrease in SKM mass was lower (OR 0.43, 95% CI 0.21–0.91, *p* < 0.05) in patients with preOP sarcopenia than in those without preOP sarcopenia. In addition, a serum albumin level of <3.5 g/dL decreased the risk of losing SKM mass (OR 0.41, 95% CI 0.19–0.87, *p* < 0.05). In patients aged >65, the risk of SAT mass reduction was lower (OR 0.53, 95% CI 0.28–0.98, *p* < 0.05).

In the multivariate logistic analysis of changes in body composition for 3 years after surgery, the risk of SAT mass reduction was increased (OR 2.78, 95% CI 1.51–5.11, *p* < 0.01) in patients with preOP sarcopenia than in those without preOP sarcopenia. In case of a recurrence after surgery, the risk of reduction in SKM mass (OR 2.78, 95% CI 1.58–4.89, *p* < 0.001), SAT mass (OR 3.76, 95% CI 2.07–6.82, *p* < 0.001), and VAT mass (OR 2.34, 95% CI 1.35–4.06, *p* < 0.01) were all significantly increased (Table 3).

### 3.4. Changes in Longitudinal Body Composition According to Each Factor

In the analysis of the percentage change in longitudinal body composition of 196 patients (118 males and 78 females) using abdominal CT images at each time point from before surgery to 36 months after surgery, regardless of each factor, SKM mass decreased the most at 3 months after surgery. SAT and VAT mass decreased the most at 3 or 6 months after surgery and then slowly recovered, but the preoperative status did not recover (Figure 3 and Figure 4).

In the repeated measures test, SKM (*p* = 0.15), SAT (*p* = 0.35), and VAT (*p* = 0.70) mass were not significantly different in the course of change between the preOP sarcopenia group and the normal group (Figure 3).

However, SKM (*p* < 0.001) and SAT (*p* < 0.05) mass, but not VAT mass (*p* = 0.87), showed a statistically significant difference in the course of change according to the recurrence (in cases of recurrence, SKM and SAT mass rapidly decreased). The difference in the percentage change of SKM between groups according to recurrence was statistically significant from 30 months postoperatively (*p* < 0.05). Although not statistically significant, there was also a difference at 24 months postoperatively (−3.10% vs. −5.56% for no recurrence vs. recurrence, *p* = 0.059) before the mean recurrence date (963.1 ± 767.8 days, approximately 31.7 months). There were no significant differences in percentage change of SKM according to recurrence postoperatively at 3 months (−6.58% vs. −7.99%, *p* = 0.34), 6 months (−5.58% vs. −5.71%, *p* = 0.92), 12 months (−4.39% vs. −4.32%, *p* = 0.96), and 18 months (−3.13% vs. −4.80%, *p* = 0.20). In addition, the percentage change of SAT showed a statistically significant difference from 36 months (*p* < 0.05) after surgery between the two groups according to recurrence (Figure 4).

## 4. Discussion

This study is the first to analyze changes in the longitudinal body composition of long-term pancreatic head cancer survivors by analyzing a large number of abdominal CT image data using AI technology. Long-term survivors did not recover to their preOP body composition state, even in cases without recurrence. In addition, the presence of preOP sarcopenia at 1 year postoperatively and recurrence at 3 years postoperatively were factors that significantly influenced the change in body composition. Furthermore, in cases with recurrence during the survival period, the SKM and SAT mass tended to decrease more significantly with time than in cases without recurrence.

SKM, SAT, and VAT mass decreased after surgery and recovered gradually, but the preoperative state did not recover in long-term survivors in this study. Depending on the type of cancer, the weight and body composition changes in survivors show differing patterns. Women who survive for >4 years after breast cancer surgery gain weight compared to women without cancer, especially when chemotherapy is administered [15]. However, in another study, the SKM mass of large B-cell lymphoma survivors decreased during treatment and recovered to baseline within 2 years, but the SAT and VAT mass increased [16]. In particular, the peak decrease in all types of body tissue analyzed was after an average of 3–6 months after surgery in this study, which is the time when cancer anorexia–cachexia syndrome severely develops postoperatively [7]. Pancreatic cancer is a representative cancer in which cachexia occurs. Cachexia is associated with low overall survival and decreased quality of life (QOL), physical function, and negative social and psychological impacts [17]. Postoperative care providing intensive nutritional support [7] and exercise intervention [8] to patients, especially 3–6 months after surgery, is necessary to prevent the aforementioned negative impact of cancer cachexia.

The percentage change in SKM mass (males vs. females; −7.0 ± 10.2% vs. −3.6 ± 9.2%) was significantly reduced in males compared to females in this study (Table 2). In previous studies, including elderly individuals living in a community, males had a higher rate of SKM reduction than females [18]. These sex differences are owing to differences in the effects of such factors as sex hormones or insulin-like growth factor-1 on muscle biology [19]. In particular, testosterone increases muscle size and strength in males [20], but estrogen, a female hormone, is not associated with muscle mass or strength [21]. Thus, as the sex hormone of each sex decreases with age, males show higher SKM loss rates than females as testosterone levels decrease. Because the decrease in SKM mass may increase the risk of deterioration of physical function [22], falls and fractures [23], and decreased QOL [24] when long-term pancreatic cancer survivors become older, male pancreatic cancer survivors should pay more attention to the decrease in SKM mass after pancreatic cancer surgery than females.

In this study, to analyze the factors affecting the change in body composition at each time point after surgery, the analysis was divided into 1 and 3 years after surgery. preOP sarcopenia was the factor that had the greatest influence on the change in body composition at 1 year after surgery. We found that SKM mass had a lower risk of reduction during survival if preOP sarcopenia was present. This is consistent with the results of a previous study by Sebastiano et al. in which cancer was detected earlier in the preOP sarcopenia group due to the prominent weight loss [5]. However, in our study, there was no statistically significant difference in factors affecting differences in SKM mass such as age (58.6 ± 8.7 vs. 60.8 ± 10.7 years, *p* = 0.08), cancer stage (*p* = 0.90), recurrence (*p* = 0.69), and adjuvant treatment (*p* = 0.12) between the sarcopenia group and normal group. These findings can be assumed that since the SKM mass of patients with preOP sarcopenia had already significantly reduced before surgery, there is not much remaining SKM mass that can be further reduced after surgery due to cancer cachexia. However, the SAT and VAT mass lipolysis was higher in patients with preOP sarcopenia (Table 3). Energy imbalance occurs due to increased energy-wasting caused by tumor metabolism and decreased energy intake due to intestinal malabsorption after surgery in the pancreatic cancer patient. Meanwhile, to resolve this energy imbalance, cancer cachexia occurs by increasing lipolysis in AT and proteolysis in SKM [25,26]. Since the preOP sarcopenia group has relatively less skeletal muscle mass that can undergo proteolysis, it is presumed that the % change in SAT and VAT is greater through more active lipolysis in the preOP sarcopenia group. Therefore, patients with preOP sarcopenia may have a higher risk of cancer cachexia than non-sarcopenic patients. They have a reduced SKM mass and a greater reduction in AT after surgery. In addition, the group with a low preOP serum albumin level (≤3.5 g/dL) had a lower risk of SKM mass reduction during survival, which can be interpreted as the same mechanism as preOP sarcopenia [27].

The factor that had the greatest influence on the change in body composition at 3 years postoperatively was disease recurrence after surgery. SKM, SAT, and VAT mass decreased significantly in the recurrence group compared to the non-recurrence group. We presume that this finding resulted from the effects of tumor factors, such as proteolysis-inducing factor [28] and lipid mobilizing factor [29] produced by relapsed tumor cells. In addition, pro-inflammatory cytokines, such as TNF-α, IL-1, and IL-6, produced in the tumor micro-environment may have also affected cachexia in the group that experienced recurrence [26]. Furthermore, the impact of surgery after recurrence and adjuvant treatments such as chemotherapy on body composition should be considered [30]. The subjects of this study had disease recurrence diagnosed on an average of 963.1 ± 767.8 days (approximately 31.7 months) after surgery. SKM mass showed a statistically significant decrease in the recurrence group from 30 months after surgery (Figure 4a). In addition, although not statistically significant, the percentage change in SKM mass in the recurrence group was lower, even at 24 months postoperatively (−3.10% vs. −5.56%, *p* = 0.059). In previous studies, pathology such as lymph node metastasis [31] or an increase in preoperative serum markers such as serum CA 19-9 levels [32] may help predict the recurrence of pancreatic cancer. In this study, since SKM mass showed a tendency to decrease to a greater extent before the average recurrence date in the recurrence group, it is necessary to perform an additional prospective study to determine whether SKM mass reduction can be used as an early detector of recurrence during the postoperative follow-up of patients.

In particular, the reduction rate of SAT mass tended to be greater than that of VAT mass in cases of recurrence in this study (Figure 4b,c). In a previous study, Batista et al. reported that the presence of a tumor increased the expression of IL-6 mRNA in SAT. Still, there was no such evidence in VAT, and the SAT is more closely related to cancer cachexia than VAT [33]. In addition, a study using SAT obtained from patients with gastrointestinal cancer found that fibrosis and inflammatory cell infiltration were significantly increased in the SAT of patients with cancer cachexia compared to patients with weight-stable cancer [34]. Previous studies investigating short-term longitudinal body composition changes in pancreatic cancer patients have reported that SAT and VAT mass decreased to a similar extent during survival [6,35], but no studies have performed subgroup analysis according to disease recurrence in long-term survivors. In this study, the decrease in SAT mass in patients with recurrence was greater than that in patients without recurrence, but there was little difference in VAT mass. This is consistent with the results of the study above by Batista et al. [33]. Clinicians should pay more attention to the possibility of a recurrence if a decrease in SAT mass is evident compared to the decrease in VAT mass during follow-up in patients after pancreatic cancer surgery.

This study has some limitations. First, the possibility of selection bias while collecting data that satisfy the strict criteria set in this study cannot be excluded. Second, since this study only included Asian individuals and was performed in a single-center, there are limitations in generalizing the results to other regions and races. Despite these limitations, this study is meaningful as it is the first study to analyze a large number of abdominal CT data using AI technology and determine changes in body composition in long-term pancreatic cancer survivors.

## 5. Conclusions

In conclusion, long-term pancreatic cancer survivors had the greatest decrease in body composition 3–6 months after surgery, which then gradually recovered; however, their preOP state did not recover, even after 3 years. The presence of preOP sarcopenia at 1 year postoperatively and disease recurrence 3 years postoperatively were important factors influencing the change in body composition of long-term survivors. Based on these results, interventions such as nutritional support and exercise at appropriate times to prevent the deterioration of body composition in pancreatic cancer survivors and paying close attention to changes in body composition during the follow-up will help improve patient prognosis.

## Figures and Tables

**Figure 1 jcm-10-03436-f001:**
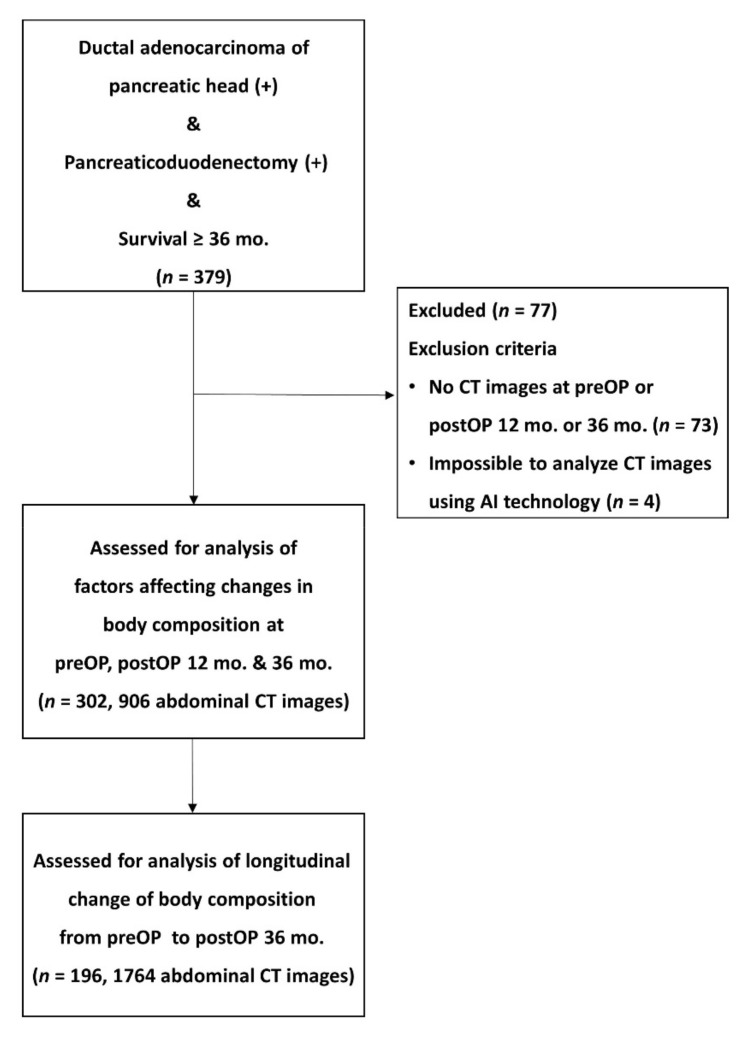
Flow diagram of study selection. CT = computed tomography.

**Figure 2 jcm-10-03436-f002:**
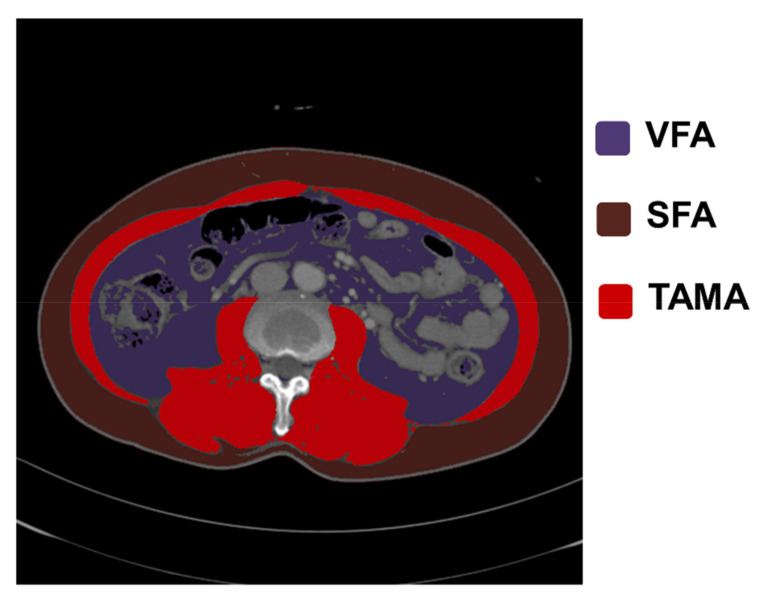
Evaluation of body composition of abdominal fat and muscle area. An axial CT image segmented into the visceral fat area (VFA), subcutaneous fat area (SFA), and total abdominal muscle area (TAMA).

**Figure 3 jcm-10-03436-f003:**
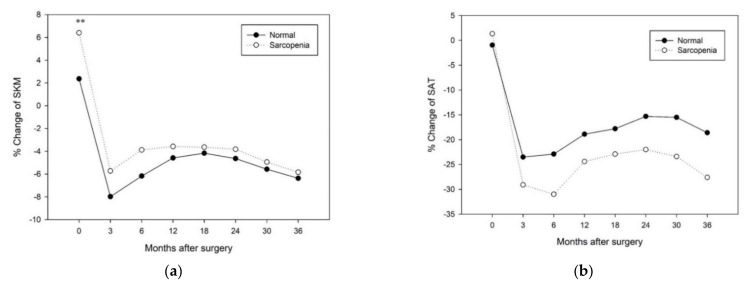
Change in longitudinal body composition according to preoperative sarcopenia: (**a**) % change of skeletal muscle mass (SKM) after surgery; (**b**) % change of subcutaneous adipose tissue (SAT) after surgery; (**c**) % change of visceral adipose tissue (VAT) after surgery. Statistical significance between groups (Student’s *t*-test; * *p* < 0.05, ** *p* < 0.01).

**Figure 4 jcm-10-03436-f004:**
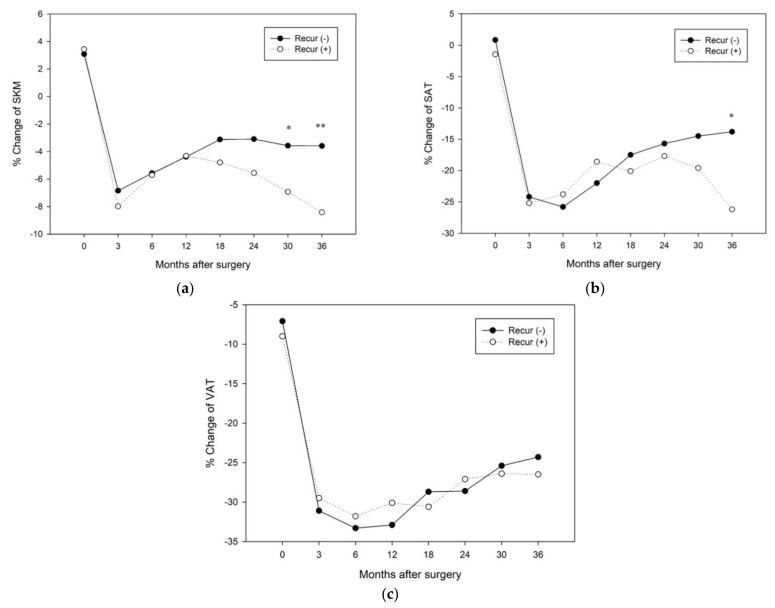
Change in longitudinal body composition according to recurrence: (**a**) % change of skeletal muscle mass (SKM) after surgery; (**b**) % change of subcutaneous adipose tissue (SAT) after surgery; (**c**) % change of visceral adipose tissue (VAT) after surgery. Statistical significance between groups (Student’s *t*-test; * *p* < 0.05, ** *p* < 0.01).

**Table 1 jcm-10-03436-t001:** General characteristics of the subjects.

Variables	Male (*n* = 181)	Female (*n* = 121)	*p* Value
Clinical information
Age (year), mean (SD)	58.9 (9.5)	59.4 (8.9)	0.68
BMI (kg/m^2^), mean (SD)	23.2 (2.3)	23.1 (2.5)	0.57
Cancer stage, *n* (%)I/II/III/IV	87/77/13/4(48.1/42.5/7.2/2.2)	65/43/8/5(53.7/35.5/6.6/4.1)	0.79

Recurrence, *n* (%)			
(+)	100 (55.2)	55 (45.5)	0.10
(−)	81 (44.8)	66 (54.5)	
Duration of recurrence after surgery (day), mean (SD)	938.5 (761.2)	1007.9 (784.8)	0.59
Adjuvant Tx, *n* (%)			
(−)	38 (21.0)	31 (25.6)	0.58
CTx only	111 (61.3)	68 (56.2)	
CTx & RTx	32 (17.7)	22 (18.2)	
Laboratory findings
Albumin (g/dL), mean (SD)	3.8 (0.4)	3.6 (0.4)	<0.05
NLR, mean (SD)	2.1 (1.2)	2.2 (2.1)	0.51
CRP (mL/dL), mean (SD)	0.93 (1.7)	1.0 (2.3)	0.72
CA 19-9 (U/mL), mean (SD)	242.9 (795.1)	260.6 (501.8)	0.83
CEA (mg/mL), mean (SD)	2.7 (2.4)	3.3 (6.9)	0.34
Body composition
SMI (cm^2^/m^2^), mean (SD)	55.1 (7.9)	44.2 (5.2)	<0.001
SFI (cm^2^/m^2^), mean (SD)	37.4 (15.2)	66.0 (21.4)	<0.001
VFI (cm^2^/m^2^), mean (SD)	39.7 (18.4)	35.4 (17.3)	<0.05
Obesity, *n* (%)	91 (50.3)	39 (32.2)	<0.01
Sarcopenia, *n* (%)	44 (24.3)	29 (24.0)	0.95

BMI, body mass index; CTx, chemotherapy; RTx, radiation therapy; Lab, laboratory; NLR, neutrophil lymphocyte ratio; CRP, C-reactive protein; CA 19-9, carbohydrate antigen 19-9; CEA, carcinoembryonic antigen; SMI, skeletal muscle index; SFI, subcutaneous fat index; VFI, visceral fat index; SD, standard deviation.

**Table 2 jcm-10-03436-t002:** Changes in body composition during the survival period.

	Male (*n* = 181)				Female (*n* = 121)			
	Pre-OP	Post-OP 36 mo	*p* Value	% Change (%)	Pre-OP	Post-OP 36 mo	*p* Value	% Change (%)
SMI (cm^2^/m^2^), mean (SD)	55.1 (7.9)	51.1 (8.6)	<0.001	−7.0 (10.2)	44.2 (5.2)	42.5 (5.9)	<0.001	−3.6 (9.2) **
SFI (cm^2^/m^2^), mean (SD)	37.4 (15.2)	29.9 (15.4)	<0.001	−6.6 (38.1)	66.0 (21.4)	51.5 (27.8)	<0.001	−22.7 (32.5)
VFI (cm^2^/m^2^), mean (SD)	39.7 (18.4)	26.9 (16.4)	<0.001	−27.5 (39.1)	35.4 (17.3)	25.9 (16.1)	<0.001	−19.9 (55.3)

OP, operation; SMI, skeletal muscle index; SFI, subcutaneous fat index; VFI, visceral fat index; SD, standard deviation. Statistical significance of % change between male and female (Student’s *t*-test; ** *p* Value < 0.01).

**Table 3 jcm-10-03436-t003:** Univariate and multivariate analysis of factors affecting changes in body composition.

	Post OP 1 Year						Post OP 3 Years					
	Univariate			Multivariate			Univariate			Multivariate		
Variables	OR	95% CI	*p* Value	OR	95% CI	*p* Value	OR	95% CI	*p* Value	OR	95% CI	*p* Value
skeletal muscle												
Age > 65 years	0.67	0.37–1.22	0.19				0.61	0.33–1.12	0.11			
Sex	0.97	0.57–1.66	0.92				1.67	0.96–2.92	0.07	1.55	0.87–2.74	0.13
Stage (I vs. others)	1.02	0.60–1.72	0.95				1.26	0.75–2.14	0.39			
Adjuvant treatment	1.95	0.96–3.96	0.06	1.67	0.81–3.44	0.17	1.72	0.87–3.42	0.12			
Recurrence	0.75	0.44–1.27	0.29				2.87	1.63–5.04	<0.001	2.78	1.58–4.89	<0.001
Body composition												
Normal vs. Sarcopenia	0.41	0.20–0.85	<0.05	0.43	0.21–0.91	<0.05	0.75	0.39–1.41	0.37			
Normal vs. Obesity	1.09	0.64–1.84	0.76				1.45	0.86–2.46	0.16			
Laboratory findings												
Albumin < 3.5 (g/dL)	0.39	0.18–0.83	<0.05	0.41	0.19–0.87	<0.05	0.66	0.34–1.28	0.21			
NLR > 4.0	0.94	0.30–2.98	1.00				0.94	0.30–2.98	1.00			
CRP > 1.0 (mL/dL)	0.98	0.47–2.06	0.97				1.22	0.58–2.58	0.60			
CA19-9 > 200 (U/mL)	0.60	0.30–1.19	0.14				0.71	0.36–1.39	0.31			
CEA > 5.0 (mg/mL)	0.71	0.29–1.71	0.44				0.64	0.25–1.62	0.34			
subcutaneous adipose tissue												
Age > 65 years	0.58	0.32–1.06	0.08	0.53	0.28–0.98	<0.05	0.67	0.37–1.22	0.19			
Sex	1.03	0.61–1.76	0.90				1.05	0.61–1.80	0.86			
Stage (I vs. others)	0.88	0.53–1.49	0.64				1.36	0.80–2.30	0.26			
Adjuvant treatment	1.42	0.74–2.74	0.29				0.90	0.48–1.66	0.73			
Recurrence	1.53	0.90–2.59	0.11				3.40	1.91–6.04	<0.001	3.76	2.07–6.82	<0.001
Body composition												
Normal vs. Sarcopenia	2.16	1.22–3.82	<0.01	2.33	1.30–4.17	<0.01	2.48	1.40–4.39	<0.01	2.78	1.51–5.11	<0.01
Normal vs. Obesity	0.71	0.42–1.21	0.21				0.87	0.51–1.49	0.62			
Laboratory findings												
Albumin < 3.5 (g/dL)	0.78	0.41–1.49	0.46				0.66	0.34–1.28	0.21			
NLR > 4.0	1.67	0.60–4.67	0.39				0.18	0.02–1.39	0.08	0.16	0.02–1.26	0.08
CRP > 1.0 (mL/dL)	0.93	0.45–1.94	0.84				1.73	0.86–3.46	0.12			
CA19-9 > 200 (U/mL)	0.81	0.43–1.55	0.52				1.10	0.59–2.06	0.77			
CEA > 5.0 (mg/mL)	0.74	0.31–1.79	0.50				0.49	0.18–1.33	0.15			
visceral adipose tissue												
Age > 65 years	0.93	0.53–1.65	0.81				0.85	0.48–1.52	0.59			
Sex	1.72	0.98–3.00	0.06	1.73	0.99–3.04	0.06	1.08	0.63–1.85	0.78			
Stage (I vs. others)	0.85	0.51–1.44	0.55				0.98	0.58–1.66	0.95			
Adjuvant treatment	1.56	0.80–3.05	0.19				1.39	0.72–2.68	0.32			
Recurrence	1.38	0.82–2.34	0.23				2.32	1.35–4.01	<0.01	2.34	1.35–4.06	<0.01
Body composition												
Normal vs. Sarcopenia	2.03	1.15–3.61	<0.05	2.05	1.15–3.65	<0.05	1.19	0.66–2.17	0.56			
Normal vs. Obesity	1.40	0.83–2.37	0.21				1.62	0.96–2.74	0.07	1.64	0.96–2.80	0.07
Laboratory findings												
Albumin < 3.5 (g/dL)	0.72	0.37–1.38	0.32				0.89	0.47–1.68	0.72			
NLR > 4.0	1.70	0.61–4.77	0.39				0.63	0.18–2.26	0.58			
CRP > 1.0 (mL/dL)	0.85	0.40–1.81	0.68				1.39	0.69–2.79	0.36			
CA19-9 > 200 (U/mL)	0.94	0.50–1.78	0.86				0.87	0.45–1.66	0.67			
CEA > 5.0 (mg/mL)	0.94	0.40–2.19	0.88				1.00	0.43–2.35	0.99			

NLR, neutrophil lymphocyte ratio; CRP, C-reactive protein; CA 19-9, carbohydrate antigen 19-9; CEA, carcinoembryonic antigen; OR, odds ratio; CI, confidence interval.

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
