# Peer review of "Longitudinal Changes in Body Composition of Long-Term Survivors of Pancreatic Head Cancer and Factors Affecting the Changes"

_jcm, 2021, doi:10.3390/jcm10153436_

Round 1

Reviewer 1 Report

I think your study is very meaningful.

It is interesting that even long survivors did not recover to their preoperative condition.

  • Please describe why did you exclude patients with pancreatic body or tail cancer.
  • "Results" and "Discussion" about Figure 4(a)

The % change of SKM at 18 and 24 months in the recurrence group may reflect a decrease of SKM in patients who have already relapsed.  If you show the % change of SKM at 3, 6, 9, 12 months before the recurrence date, it could be clear the decrease of SKM would be a predictive marker of recurrence.

  • Figure 3 and 4

The % change at 0 months is not “0”. I didn’t understand why?

Reviewer 2 Report

This manuscript by Kong et al describes changes in body composition of long-term survivors of pancreatic head cancer. Only few studies have addressed this topic. This is an interesting and well-designed study; however, we have the following concerns regarding the manuscript.

(Major)

The authors limited indications to only pancreatic head cancer cases. The authors should provide an explanation as to why cases of pancreatic body and tail cancer were not included.

The authors demonstrated that decreases in body composition failed to recover even 3 years after surgery. There may be several reasons why body composition did not recover in the present study, even in cases without recurrence.

One of the factors underlying decreasing body composition may be tumor recurrence. The reviewer suggests that the recurrence may cause decreasing body composition and may confound the results of the present study. Accordingly, this finding may not identify cases of pancreatic cancer with a good prognosis.

In figure 3, cases with sarcopenia had decreased subcutaneous adipose tissue (SAT) and visceral adipose tissue (VAT) compared to skeletal muscle (SKM). Although adipose tissues in cases with sarcopenia had small volumes, the authors should describe potential reasons for the differences observed between sarcopenic and normal cases.
